# Narrow Precursor Mass Range for DIA–MS Enhances Protein Identification and Quantification in Arabidopsis

**DOI:** 10.3390/life11090982

**Published:** 2021-09-18

**Authors:** Huoming Zhang, Dalila Bensaddek

**Affiliations:** Core Labs, King Abdullah University of Science and Technology, Thuwal 23500-6900, Saudi Arabia; huoming.zhang@kaust.edu.sa

**Keywords:** data-independent acquisition (DIA), mass spectrometry, precursor mass range selection, Arabidopsis, quantitative proteomics

## Abstract

Data independent acquisition–mass spectrometry (DIA–MS) is becoming widely utilised for robust and accurate quantification of samples in quantitative proteomics. Here, we describe the systematic evaluation of the effects of DIA precursor mass range on total protein identification and quantification. We show that a narrow mass range of precursors (~250 m/z) for DIA–MS enables a higher number of protein identifications. Subsequent application of DIA with narrow precursor range (from 400 to 650 m/z) on an Arabidopsis sample with spike-in known proteins identified 34.7% more proteins than in conventional DIA (cDIA) with a wide precursor range of 400–1200 m/z. When combining several DIA–MS analyses with narrow precursor ranges (i.e., 400–650, 650–900 and 900–1200 m/z), we were able to quantify 10,099 protein groups with a median coefficient of variation of <6%. These findings represent a 54.7% increase in the number of proteins quantified than with cDIA analysis. This is particularly important for low abundance proteins, as exemplified by the six-protein mix spike-in. In cDIA only five out of the six-protein mix were quantified while our approach allowed accurate quantitation of all six proteins.

## 1. Introduction

Mass spectrometry (MS)-based proteomics has become a method of choice to study complex biological systems in the last decade [1]. Quantitative proteomics strategies that can simultaneously identify and quantify a large number of proteins not only contribute to our understanding of biological systems at the molecular level but also can be used in biomarker discovery [2,3] and different stages in the drug discovery process [4,5,6]. Common quantitative proteomics strategies include isotopic labeling [7,8,9] and label-free approaches [1,10]. In these approaches, data dependent acquisition (DDA) is often used to isolate and fragment peptides of top-ranked intensity (e.g., top 20) as they elute from the liquid chromatography (LC) separation column directly into the MS sampling region. The peptide ions are selected for fragmentation using isolation windows of less than two Thomson depending on the instrument capabilities. Such methods are generally high-throughput but suffer from limited reproducibility of measurement, especially for peptides of relatively low abundance. This is due to the nature of DDA experiments, where peptide selection is largely stochastic. In addition, these experiments often require off-line fractionation prior to online LC-MS analysis in order to achieve an in-depth proteomics coverage, thus increasing the instrument time requirement. This, coupled with the expensive isotopic reagents for multiplexing in the case of TMT/iTRAQ, makes DDA cost ineffective, not to mention the issue of ratio compression in which the experimentally measured fold change is smaller than the anticipated or actual fold change between samples, especially for protein expression with a relatively large fold change [11,12]. In the case of label-free DDA analyses, it is very important to control the LC reproducibility especially for large numbers of samples. Targeted quantitative proteomics [12,13], on the other hand, measures a small number of known proteins/peptides with higher sensitivity and reproducibility but it has limited use for proteome-wide survey.

Data-independent acquisition (DIA/SWATH) MS-based proteomics was developed more recently with the advantages of high reproducibility and accurate quantitation in a high-throughput fashion [14]. In contrast to DDA-MS, DIA–MS cyclically fragments all peptides across the entire precursor mass range (typically 400–1200 m/z). In each cycle, a large isolation window (e.g., 25 m/z units) is used to sequentially isolate peptides from the defined precursor mass range to be fragmented. Typically, this is achieved using 32 windows and 2–3 s cycling time in most current mass spectrometers. Since a peak normally elutes from the LC within 15–30 s, a total of 5–10 MS data points can be collected and a peak profile can be well constructed for each peptide, thus enabling accurate peptide quantitation. It should be noted that DIA isolates and fragments a group of precursors using a wide window, thereby generating highly complex fragment ion spectra from a complex mixture (e.g., tissue or cell lysate). Subsequent targeted extraction of individual peptide spectra can be performed by matching with a preconstructed ion library [15,16], an in silico library [17] or using a library-free approach [18].

The sensitivity of DIA–MS relies heavily on the fragment ion spectral quality, with higher resolution and less complexity of spectra significantly enhancing identifications. Significant improvements were reported by Bruderer et al. [19]; the authors combined high resolution acquisition in both MS1 and MS2, high sample loading with indexed retention time (iRT) peptides spike-in for library generation along with improved algorithm for data analysis to quantify ~7100 proteins from HEK-293 cells and >8000 proteins from a mouse brain tissue using single shot DIA–MS. Similarly, reducing the complexity of DIA spectra through a variable-window isolation strategy to ensure a similar number of precursors are isolated for each DIA fragmentation window across the defined precursor mass range was also shown to enhance protein identification by about 10% [20]. Amodei et al. used overlapping DIA windows, which significantly improved the specificity of fragment ions and thus the sensitivity in detection [21]. Borras et al. developed a DIA+, which combined signals from identical peptides with different charge states through multiplexing scans. This resulted in an improved signal-to-noise, an additional number of fragments and a reduction in the number of analytes per composite window, thereby increasing the identification and quantification of peptides in complex samples [22]. More recently, Cai et al. reported a PulseDIA method in which they combined several DIA–MS analyses for the same sample with complementary narrow DIA isolation windows. As such, they were able to identify 29% more proteins [23]. Furthermore, using a small sample specific library was shown to reduce the false discovery rate (FDR) in the spectral match and improve in protein identification and quantification [24,25]. Certainly, with advances in MS technology, either with faster scan rates and incorporation of additional separation dimensions such as ion mobility, we would expect to be able to quantify high numbers of proteins in a single shot. For example, using a timsTOF pro in diaPASEF mode enabled the quantitation of more than 7000 proteins in 120 min gradient [26], while FAIMS Orbitrap was used to quantify over 8000 proteins with 6 h gradient [27]. 

Arabidopsis is one of the most important model organisms. It is the first plant to have its genome fully sequenced. It is a widely used a model organism for plant research. Earlier, we created a comprehensive, high quality Arabidopsis mass spectral library that is useful for DIA–MS proteomics quantitation [16]. Since the majority of peptides in the library fall within a narrow mass range (e.g., ~60% in 400–700 m/z), we hypothesized that using a small precursor range for DIA–MS would result in an increase in resolution, ion specificity and match against small spectral library which in turn would increase overall identification without compromising protein quantitation accuracy. To this end, we systematically evaluated the effect of precursor mass range on protein identification and quantitation accuracy using Arabidopsis samples. We show here that it is possible to obtain a 34.7% increase in the number of protein groups quantified from a single injection with narrow precursor mass range (with size of 250 m/z units) and a 54.7% increase when combing three DIA–MS analyses with narrow precursor mass ranges obtained using gas-phase fractionation.

## 2. Materials and Methods

### 2.1. Arabidopsis Root Cell Suspension Culture

Cells isolated from roots of Arabidopsis were grown in Gamborg’s B5 basal salt mixture (Sigma-Aldrich, St. Louis, MO, USA) with 2,4-dichlorophenoxyacetic acid (2,4-D; 1 mg mL^−1^) and kinetin (0.05 μg mL^−1^) in sterile flask as described [28]. The cells were harvested by draining off the media using a Stericup^®^ filter unit (Millipore, Billerica, MA, USA) and immediately flash frozen in liquid nitrogen and stored at −80 °C until use.

### 2.2. Protein Extraction and Digestion

The plant cells were ground in liquid nitrogen with a prechilled mortar and a pestle. The fine powder was resuspended in the extraction buffer (50 mm Tris, pH 8, 8 M urea and 0.5% SDS) supplemented with protease inhibitor (Roche Diagnostics GmbH, Mannheim, Germany) and homogenized with a Dounce homogenizer. Subsequently, the crude homogenate was subjected to 30 cyclic high/low pressurization (50 sec of 35,000 PSI and 10 sec of ambient pressure) using a pressure cycling technology (Barocycle, PressureBioSciences, MA, USA). The extracts were then centrifuged at 10,000 *g* for 5 min at 4 °C. The proteins in the supernatant were purified using methanol/chloroform precipitation and dried under vacuum. The dried pellets were dissolved into the extraction buffer with the aid of sonication. The protein content was determined using a microBCA kit (Thermo Scientific, Waltham, MA, USA). Approximately 100 µg of proteins were reduced, alkylated and digested with trypsin using a FASP method [29]. The digests were desalted prior to LC-MS analysis with microcolumns packed with C18 material. To prepare protein digests from 6 proteins (Glycoprotein standards set, Thermo Scientific) for spike-in experiment, 10 µg of each protein was dissolved into 50 mM ammonium bicarbonate, reduced and alkylated prior to trypsin digestion as described [30].

### 2.3. Mass Spectrometric Analysis Using Data Independent Acquisition (DIA) Mode

The DIA–MS analysis was conducted using an Orbitrap Fusion Lumos mass spectrometer (Thermo Scientific) coupled with an UltiMate^TM^ 3000 UHPLC (Thermo Scientific). Approximately 0.8 µg of peptide mixture was injected into a precolumn (Acclaim PepMap, 300 µm × 5 mm, 5 µm particle size) and desalted for 15 min with 0.1% FA in water at a flow rate of 5 µL/min. The peptides were eluted into an EasySpray C18 column (50 cm × 75 µm ID, PepMap C18, 2 µm particles, 100 Å pore size, Thermo Scientific) and separated with a 130-min gradient at constant 300 nL/min, at 40 °C. The gradient was established using mobile phase A (0.1% FA in H_2_O) and mobile phase B (0.1% FA, 95% ACN in H_2_O): 2.1–5.3% B for 5 min, 5.3–10.5% for 15 min, 10.5–21.1% for 70 min, 21.1–31.6% B for 18 min, ramping from 31.6 to 94.7% B in 2 min, maintaining at 94.7% for 5 min, and 4.7% B for 15-min column conditioning. The sample was introduced into the Fusion Lumos through an EasySpray (Thermo Scientific) with an electrospray potential of 1.9 kV. The ion transfer tube temperature was set at 270 °C. The MS parameters included application mode as standard for peptide, RF lens as 30%, default charge state of 3 and the use of EASY-IC as internal mass calibration in both precursor ions (MS1) and fragment ions (MS2). Other parameters for MS1 include a resolution of 60,000 (at 200 m/z), a maximum ion accumulation time of 50 milliseconds, a target value of 2e5, and data type of profile. For the MS2 by DIA, the mass defect was 0.9995. The normalized HCD collision energy was set to 30% for peptide fragmentation. MS2 spectra were recorded in centroid mode at 30,000 resolution from 350–1500 m/z, with an AGC target of 1e6 and a maximum ion accumulation time of 100 ms. To determine an optimal precursor window size for quantifying a higher number of protein groups, the sizes of precursor mass range from 150 to 800 (m/z unit) (Figure 1a) were evaluated for DIA–MS analysis. Accordingly, a quadrupole isolation window size was set from 4–20 (m/z unit) for each defined precursor mass range in order to achieve a similar number of total scan events (*n* = 39–41) per cycle. 

In order to compare the identification and quantification between a conventional DIA (cDIA) and the DIA with optimal precursor scanning range, the 6 standard protein digests were spiked into the 1 µg Arabidopsis peptides as detailed in Table 1. In cDIA, a precursor mass ranges from 400 to 1200 (m/z), with an isolation window of 25 m/z and a total of 32 scan events. For the DIA with the optimal precursor scanning range as determined as above, we combined DIA with a gas-phased fractionation (GPF) strategy which divides precursor mass-to-charge ranges into several smaller precursor mass ranges for DIA–MS analysis. As such three independent DIA–MS experiments were performed for the same sample with precursor mass ranges 400–650; 650–900 and 900–1200 m/z, respectively. The isolation windows were 8 (for precursors 400–650 and 650–900 m/z) and 9 m/z (for 900–1200 m/z) which resulted in a total of 32 and 31 scan events per cycle accordingly. All other parameters were the same as aforementioned. Each experiment was conducted in triplicate.

### 2.4. DIA–MS Data Analysis Using Spectronaut

#### 2.4.1. Library Generation

All DIA–MS data files and the search archive of our published Arabidopsis library [16] were combined for library generation using Spectronaut Pulsar (version 14.10.201222.47784, Biognosys, Zurich, Switzerland). The protein database was the combination of a TAIR11 proteome sequence and the six spike-in sequences. The default settings for database match include: full specificity trypsin digestion, peptide length of between 7 and 52 amino acids and maximum missed cleavage of 2. Besides, lysine and arginine (KR) were used as special amino acids for decoy generation, and N-terminal methionine was removed during preprocessing of the protein database. Carbamidomethylation at cysteine was used as a fixed modification, protein N-terminal acetylation and methionine oxidation were set as variable modifications. The false discovery rates (FDRs) were set as 0.01 for the peptide-spectrum match (PSM), peptide and protein identification. The other Biognosys default spectral library filters include amino acid length of ion more than 2, ion mass-to-charge between 300 and 1800 Da and minimum relative intensity of 5%. The best 3–6 fragments per peptide were included in the library. The iRT calibration was required with minimum R-Square of 0.8.

#### 2.4.2. Quantitation Analysis

The DIA data were analyzed by Spectronaut using both a library-based matching and directDIA approach. In the directDIA, a collection of pseudo-MS2 spectra was generated from the DIA data and directly used for database search and library construction. The resulting DIA data specific library was then used for targeted analyses of the same DIA data. In the library approach, the newly generated spectral library from Section 2.4.1 was used for spectral matching. In both approaches, data extraction was based on maximum intensity in both MS1 and MS2 spectra with relative mass tolerances of 10 and 20 ppm, respectively. Retention time window of Extracted ion Chromatogram (XIC) was set dynamically but with a correction factor of 0.5 to enhance specificity. The parameters in the calibration category include allowing source specific iRT calibration at automatic mode, precision iRT with local nonlinear regression and a mass tolerance of 40 ppm for both MS1 and MS2 spectra. The Biognosys default settings were applied for both identification and quantification. These parameters were excluding duplicate assay; generation decoy based on mutated method at 10% of library size, and estimation of FDRs using q-value as 0.01 for both precursors and proteins. The *p*-value was calculated by the kernel-density estimator. Interference correction was activated and a minimum of 3 fragment ions and 2 precursor ions were kept for the quantitation. The area of extracted ion chromatogram (XIC) at MS2 level was used for quantitation. Peptide (stripped sequence) quantity was measured by the mean of 1–3 best precursors, and protein quantity was calculated accordingly by the mean of 1–3 best peptides. Local normalization strategy and q-value sparse selection were used for cross run normalization. To determine the differential abundance between samples, the major group (quantification settings) was used for the differential abundance grouping, and the precursor ion (quantification settings) was chosen in the smallest quantitative unit. In addition, an unpaired t-test with assuming equal variance, group-wise testing correction and clustering was carried out. Proteins with a fold-change of higher than 1.5 and a q-value of less than 0.001 were considered as differentially expressed proteins.

## 3. Results

### 3.1. Effects of DIA Precursor Mass Range Selection on the Protein Identification

In the Arabidopsis library, the precursors are distributed along the mass-to-charge range (m/z) of 375–1400 with approximately 60% of precursors concentrated within the narrow region of 400–700 m/z. To systematically evaluate the effects of the size of precursor mass range on the number of protein identifications, we performed DIA analyses with various sizes of precursor mass range from 150 to 800 with 50 m/z increments. The precursor scan range started from 550–700 m/z (150 m/z units) to 400–1200 m/z (800 m/z units) as shown in Figure 1a.

Each DIA–MS file was analyzed using Spectronaut software. Precursors, peptides and proteins with a *Q*-value <0.01 were considered as positive identification and used for statistical analyses. As can be seen in Figure 1, the total number of identified precursors from each DIA–MS analysis, initially, increased with the size of precursor scan range then plateaued at the size of 400 (Figure 1b). A similar trend was observed with the identified peptides but it reached a steady level at a slightly smaller precursor scanning size (300–350) (Figure 1c). In contrast, the highest number of protein groups (*n* > 8,250) was identified from 150–300 precursor scanning m/z. The number dropped by ~30% when wider precursor scanning ranges (700–800 m/z units) were included in DIA–MS (Figure 1d).

### 3.2. Narrow Precursor Scan Range Quantified Higher Number of Proteins

Having established an optimal size of precursor mass range of 200–300 (m/z), we combined DIA experiments with gas phase fractionation (GPF), where a wide precursor range used in conventional DIA (cDIA: 400–1200 m/z) was divided into precursor scan ranges: 400–650, 650–900 and 900–1200 m/z for three independent DIA–MS experiments respectively. Consistent with the fact that although narrow precursor scan range resulted in relatively lower peptide identifications (Figure 1c and Figure 2c), both narrow precursor range GPF-DIA analyses (with 400–650 and 650–900 m/z, respectively) resulted in the identification of higher numbers of protein groups (*n* = 8798 and 8581, respectively) compared to the cDIA (*n* = 6530) with 34.7 and 31.4% increase, respectively, whereas the GPF 900–1200 (*n* = 6344) identified similar number of protein groups (Figure 2a, Appendix A) despite a much lower number of precursors in this mass range. The ratios of common identifications over total proteins from the triplicate analysis were all above 88%, with highest of 97% in cDIA method. The missing values of protein groups (defined as not consistently identified in all replicate analyses) slightly increased in GPF-DIA likely due to the contribution from much lower abundance proteins. Overall, our data showed high reproducibility in identifying proteins across the full dynamic range for 8798 proteins from our GPF-DIA.

The benefit of using narrow precursor windows becomes more obvious when combining the three GPF-DIA runs for analysis (Figure 2b). A total of 10,099 protein groups were identified from combined GPF-DIA (cGPF-DIA, Appendix A) when matching to the Arabidopsis spectral library, which represents an approximate increase of 54.7% compared to the cDIA approach (*n* = 6530, Appendix A). Similarly, the numbers of peptides increased by 41.6% in cGPF-DIA (Figure 2d). When using a directDIA approach to analyze the data, we observed an apparent increase of ~45.3 and 82.3% in the number of protein groups and peptides quantified, respectively in cGPF-DIA compared to the cDIA data. This substantial increase (82.3%) in peptide identification in directDIA analysis illustrates the importance of the high-quality MS2 spectra for DIA analysis, in particular in the absence of a comprehensive spectral library. Use of narrow precursor mass range DIA–MS enhanced MS spectra quality and thus greatly improved the peptide identification in the directDIA analysis. It is also noteworthy that matching DIA data with a spectral library generally resulted in 5–10% more protein groups than directDIA where no preconstructed library was used.

### 3.3. cGPF-DIA Recovered Low Abundance of Proteins

To investigate the relative abundances of identified proteins from matching Arabidopsis library, the protein intensities were log transformed with a base of 10, and plotted using histograms. Similar distributions of common identifications were observed in conventional DIA and cGPF-DIA (Figure 3a). When all identified proteins were used for the plot, the distribution of protein abundances was clearly skewed to the relatively low abundance side (Figure 3b). This suggests that the increase in protein measurement observed in the cGPF-DIA method consists mainly of low abundance proteins. As a result, the dynamic range of protein identifications in cGPF-DIA was increased by ~1 order of magnitude mainly in the lower abundance range to span nearly 6 orders of magnitude. This demonstrates that a higher sensitivity can be achieved by using narrow precursor mass range DIA–MS.

### 3.4. Narrow Precursor Ranges DIA Enabled Accurate Quantitation

To assess quantitative accuracy of the workflow, every experiment was carried out three times. The median coefficient of variation (CV) of protein group quantities was ≤6.5% in cDIA and ≤5.9% in narrow precursor range DIA, respectively. In particular, they were lower than 4.3% for the DIA with a narrow size (250 m/z units) of precursor range, indicating higher quantitation accuracy as compared to that in cDIA.

Next, we performed a direct comparison between the cDIA and cGPF-DIA. First, we extracted common identifications (*n* = 6306) and plotted their intensities against each other (Figure 4). The correlation coefficient was >0.9 illustrating the overall similarity in quantitation achieved by the two methods. Second, the robustness of identifying differential expressions from the complex samples was demonstrated using protein mixtures containing the same amount of Arabidopsis samples (1 µg) but different amounts of 6-protein mix spike-in (Table 1). As expected, there was no Arabidopsis protein identified as significantly changing in either approach, suggesting a highly accurate and reliable measurement by data independent acquisition (Figure 5). The six spike-in proteins were all identified in both approaches; however, they were successfully quantified with accurate ratios and high confidence (*Q*-value < 0.001) only in cGPF-DIA (Figure 5b). The detailed information about their identification and quantitation are presented in Appendix A (protein information), Appendix A (peptide information) and illustrated in Appendix A. One of the proteins, namely FETUB, was not found to have a significant difference due to the high *Q*-value (*Q*-value = 0.12) in cDIA approach (Figure 5a, Table 1). In addition, the measured ratio of FETUB from the measurement of two samples by cDIA was rather inaccurate, 19.7 instead of 5 (initial mix ratio). In general, most proteins including Arabidopsis proteins were quantified at lower *Q*-value and thus with higher confidence from our narrow precursor mass range DIA.

## 4. Discussion

Through systematic assessment of the effect of precursor mass range on the extent of protein identification and quantitative accuracy, we reported a DIA workflow of using a narrow precursor mass range that outperformed conventional DIA with a wider precursor mass range. Our method successfully identified a 34.7% higher number of protein groups with additional proteins mostly at a lower abundance range using a single injection for DIA–MS of narrow precursor mass range (400–650 m/z). When combining three independent DIA–MS analyses of different narrow precursor mass range by gas-phase fractionation, a 54.7% higher number of protein groups (*n* = 10,099) could be obtained as compared to the conventional DIA approach. This represents approximately 65% of our comprehensive Arabidopsis spectral library obtained using extensive DDA runs [16]. Higher numbers of protein identification were largely attributed to the increase in spectral resolution and specificity. When using narrow precursor mass range, the isolation window for DIA is smaller and contains fewer precursor ions while the AGC target remains the same meaning that for each precursor in the range more ions can be accumulated, resulting in spectra with better signal-to-noise ratios.

Furthermore, the robustness and advantages were demonstrated for identifying differential expression in a highly complex mixture. Instead of spiking in a completely different proteome as in [19], we introduced only a few proteins in relatively low amounts into Arabidopsis root cell digest. This is to mimic real biological events where the majority of proteins remain unchanged while only a small number of proteins would respond to a stimulus. As expected, using our narrow precursor mass range DIA, all proteins were accurately quantified with high confidence while there is no falsely quantified target. In contrast, conventional DIA quantified five out of the six spiked-in proteins and failed to identify FETUB, which had a more extreme ratio (r = 19.7) and high *Q*-value (*Q* = 0.12).

Importantly, the workflow is simple to perform, and uses the same data analysis approach as conventional DIA. In theory, inclusion of a wide precursor m/z (400–1200) window in DIA would enable the acquisition of fragment spectra for every precursor within the defined mass range, and thus identify a near complete proteome. However, in practice, due to the limited mass spectrometry sensitivity and high dynamic range of a typical proteome, the wide precursor mass ranges usually result in chimeric spectra where low abundance ion fragments are suppressed, which prevents their identification during spectral matching. Although this issue has been long recognized and many attempts were made to address it, limited success has been achieved using overlap DIA [21,22,31]. In addition, such approaches often require the implementation of a new data analysis algorithm, thus limiting its use in the research to community.

Since the early days of DIA, a high-quality spectral library either constructed experimentally or computationally has been considered a prerequisite for good spectral matching in the DIA analysis. As such, a lot of effort has been put towards building a comprehensive library via extensive DDA analyses [15,16,32,33]. There are various studies showing that a sample specific library results in better identification [34,35], thus justifying the resources required for library generation. In 2017, Biognosys introduced a library-free DIA approach (coined as directDIA) which could identify up to 90% of protein groups compared to a library-based approach. Our data comparing directDIA and spectral library confirmed that an additional 5–10% increase in protein identification can be obtained in the library-based approach.

In conclusion, which DIA regimen to use will always depend on the objective of the study, the resources available and cost-benefit assessment. However, our study provides few guidelines on how to improve the number of proteins quantified in a given analysis. For example, we have shown that DIA with narrow precursor mass ranges results in an increase in quantitative accuracy while improving the protein sampling, especially at the low abundance intensity range, when compared to conventional DIA. When combining three narrow precursor ranges, covering the same range as the cDIA, we were able to maximize the number of proteins quantified, while increasing the sequence coverage of these proteins. This is particularly important in the study of protein isoforms, for example as a result of splicing. Furthermore, being able to access the low abundance range is very important to elucidate signaling mechanisms in various studies.

## Figures and Tables

**Figure 1 life-11-00982-f001:**
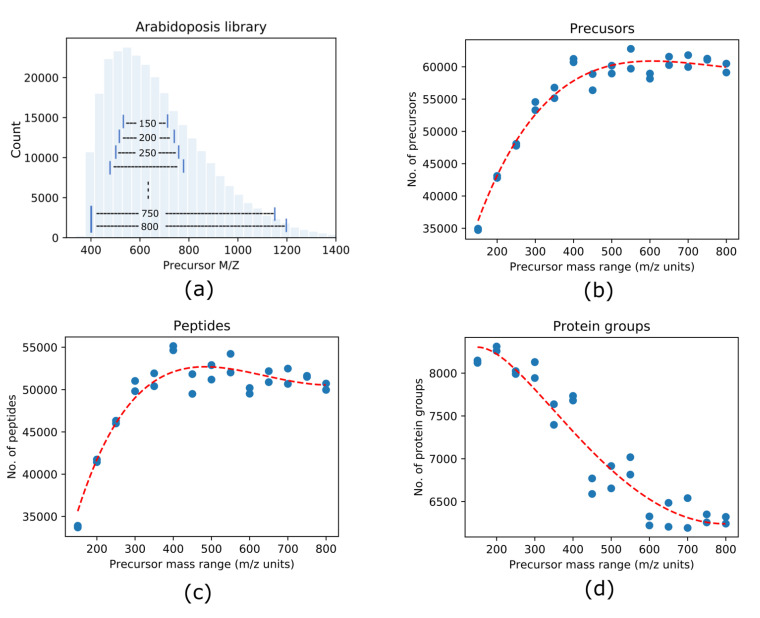
Size of precursor mass range affects DIA identification. (**a**) In the Arabidopsis precursor library, most of the precursors are distributed across the mass-to-charge range of 400–700 (~62%). The size of precursor m/z ranges from 150 (m/z) to 800 (m/z) with a step-up of 50 (m/z) was systematically assessed for DIA–MS analysis. The experiment was conducted in duplicate. (**b**) The scatter plot shows that the number of precursors identified increases with the size of the precursor mass range selection. (**c**) The scatter plot shows that the number of identified peptides increases with the size of the precursor mass range selection. (**d**) The scatter plot shows that the number of protein groups identified decreases with size of the precursor mass range selection.

**Figure 2 life-11-00982-f002:**
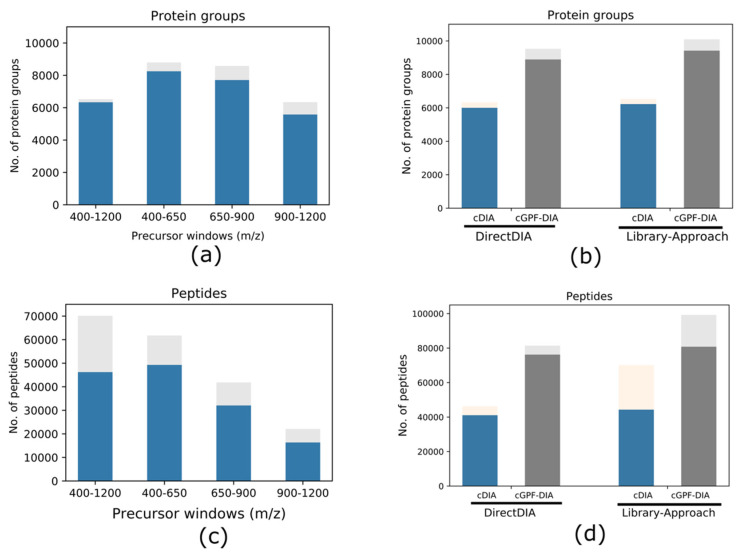
Protein groups and peptides identified from different approaches with each condition run in triplicate. The total number of protein groups (**a**) and peptides (**c**) identified from DIA–MS analyses of four different precursor mass ranges, respectively (cDIA: 400–1200 m/z, and three individual gas-phase fractionation (GPF)) by matching spectral library. Lower blue bar indicates common identification from the triplicate experiment, and upper gray bar shows the identification from 1 or 2 replicates only; whereas the total number of protein groups (**b**) and peptides (**d**) identified from cDIA and the combination of three GPF-DIA (cGPF-DIA) via directDIA (left panel) and library-based approaches (right panel), respectively. cGPF-DIA identified >45% more proteins and >40% more peptides than cDIA in both directDIA and library-based data analysis approaches. Library-based approach obtained a modest increase in protein identification compared to directDIA. Lower dark color segment indicates common identification from the triplicate analysis whereas the upper light color bar shows the identification from 1 or 2 replicates only.

**Figure 3 life-11-00982-f003:**
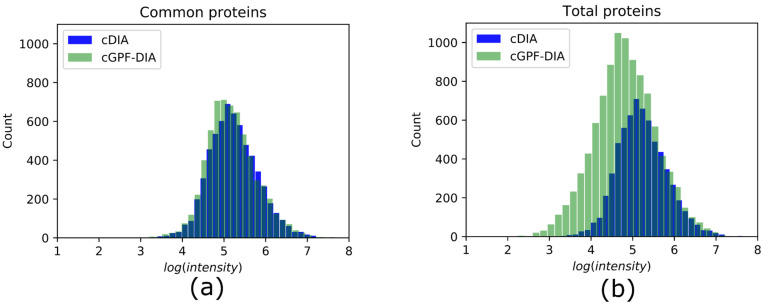
Histograms of the distribution of protein abundances measured in DIA experiments. (**a**) Distributions of common proteins identified in both cDIA (blue) and cGPF-DIA (green). Bin size is 30. The two histograms are nearly overlapping suggesting similar dynamic ranges. Most proteins have abundances from 3.2 to 7.3 (*log_10_(intensity)*). (**b**) Distributions of total protein identifications from cDIA and cGPF-DIA; the distribution of proteins in cGPF-DIA (shown in green) ranges from 2.6 to 7.3 with a clear shift to lower abundance range compared to cDIA (shown in blue) and fully encompasses the cDIA range.

**Figure 4 life-11-00982-f004:**
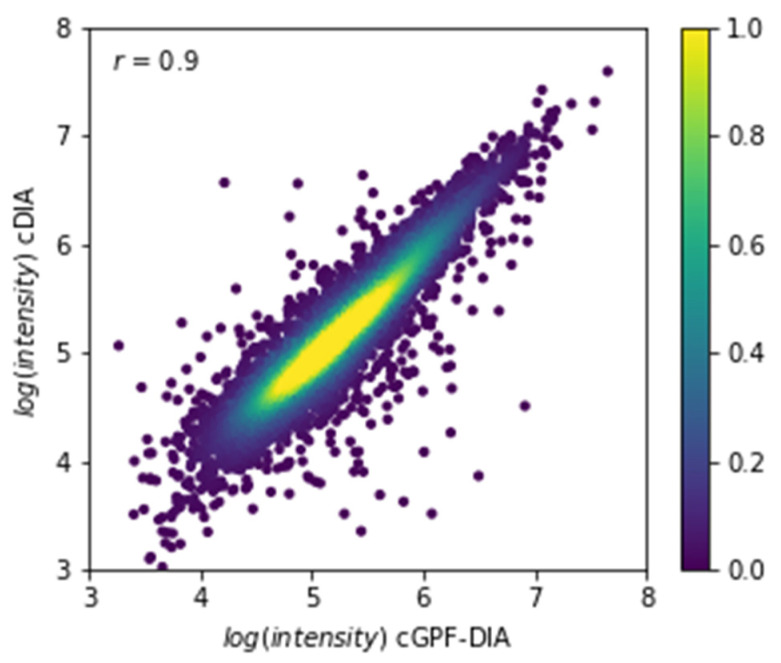
Correlation of quantitative intensities of common identifications (*n* = 6306) between cDIA and cGPF-DIA. The correlation coefficient was 0.9, and the color bar illustrates the protein intensity correlation of two representative methods.

**Figure 5 life-11-00982-f005:**
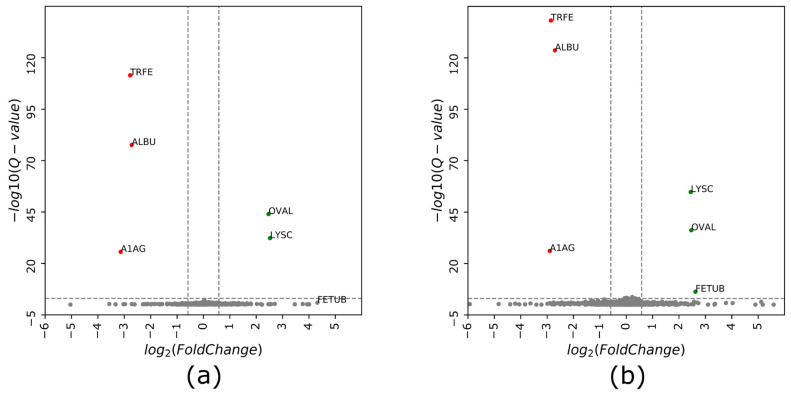
Volcano plot of protein intensity ratios log_2_ (samples 2/1) obtained using (**a**) cDIA and (**b**) cGPF-DIA approaches. Proteins with a fold change (or ratio) ≥1.5 and a *Q*-value ≤ 0.001 were considered to have a significantly different expression/spike-in ratio. Shown in red are proteins that are down-regulated while green denotes up-regulated proteins. FETUB was not identified as significant differential expression in cDIA–MS due to its relatively large *Q*-value (0.12). Sample loading amounts and measured ratios of spike-in proteins are shown in Table 1.

**Table 1 life-11-00982-t001:** Six standard protein digests were spiked into an equal amount of Arabidopsis peptide mixture with two different ratios to get sample 1 (S1) and sample 2 (S2).

Protein Information			Quantity (fmol) and Ratio of Spike-In			Quantitation by cDIA*		Quantitation by cGPF-DIA**	
	SwissProt Accession	Molecular Weight (Da)	S1	S2	S1:S2	S1:S2	*Q*-value	S1:S2	*Q*-value
BSA	P02769	69300	64.1	8.0	8:1	6.6: 1	2.5 × 10^−18^	6.5: 1	2.4 × 10^−12^
Transferrin	P02787	77070	57.7	7.2	8:1	6.8: 1	3.3 × 10^−112^	7.2: 1	7.0 × 10^−13^
alpha-acid glycoprotein	Q3SZR3	23190	191.7	24.0	8:1	8.8: 1	2.4 × 10^−26^	7.5: 1	7.7 × 10^−27^
Lysozyme	P00689	16240	34.2	171.0	1:5	1: 5.8	4.7 × 10^−33^	1: 5.4	1.7 × 10^−55^
Fetuin	Q58D62	42670	13.0	65.1	1:5	1: 19.7	1.2 × 10^−1^	1: 6.1	4.9 × 10^−7^
Ovalbumin	P01012	42890	13.0	64.8	1:5	1: 5.5	9.7 × 10^−78^	1: 5.5	6.6 × 10^−37^

* cDIA: conventional DIA; ** cGPF-DIA: combined gas-phase fractionation DIA. *Q*-value is a *p*-value that has been adjusted for the False Discovery Rate (FDR) and a *Q*-value of 0.001 is the equivalent to an FDR cutoff of 0.1% for measuring significant quantitative change.

## Data Availability

The mass spectrometry proteomics data together with the analyses using Spectronaut software have been deposited to the ProteomeXchange Consortium via the PRIDE [1] partner repository with the dataset identifier PXD026965.

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
