# Peer review of "Narrow Precursor Mass Range for DIA–MS Enhances Protein Identification and Quantification in Arabidopsis"

_life, 2021, doi:10.3390/life11090982_

Round 1

Reviewer 1 Report

In the manuscript, the authors performed an evaluation of DIA precursor mass range on Arabidopsis samples for identification and counting of proteins. Demonstrating that narrowing precursor mass range improves protein analysis in Arabidopsis is a novelty of this work. The significance of this work however needs to be further motivated.

Since the authors used only one model set of samples, i.e. Arabidopsis, they should mention “Arabidopsis” in the title.

Introduction

Line 42. “ratio compression” needs an explanation.

It should be motivated in the introduction why specifically Arabidopsis was used in this study.

A recent reference “Xue et al., 2021, Journal of Proteome Research, PulseDIA: Data-Independent Acquisition Mass Spectrometry Using Multi-Injection Pulsed Gas-Phase Fractionation” should be referred to.

Methods

Line 137. “HCD collision energy was set to 30%” – 30% of what?

Lines 149-154. “DIA was performed with precursor mass ranges 400-650; 650-900 and 900-1200 m/z respectively, using gas-phase fractionation (GPF).” – Was it 3 different injections for the different precursor mass ranges 400-650; 650-900 and 900-1200 m/z? GPF requires more explanation.

Line 177. The "directDIA" method needs more explanation. How it was performed and how was the data processed?

Line 184. A mass tolerance of 40 ppm seems to be high. Could it generate false positives?  

Table 1.

On what basis were the spike-in concentrations and ratios selected?

Describe Q-value

For Ovalbumin, ratio of spike-in shows “1:05”. Should it not be 1:5?

Results

The authors under Methods say a mass tolerance of 40 ppm was used for both MS1 and MS2 spectra, and that area of XIC as MS2 level were used for quantification. Was there disturbing background noise and how were the peak shapes? Some representative chromatograms (MS1 and MS2) and spectra of precursors, peptides and protein groups with selected examples should be given either in main article or as supplementary information.

3.1. It should be made clear which peptides the authors are referring to by giving some examples/names of detected peptides, and how were they identified and counted.

Similarly, for the protein groups – how were they identified and counted needs clarification.

3.2. “conventional DIA (cDIA: 400-1200 m/z) was divided into three precursor scan range: 400-650, 650-900, and 900-1200 m/z.” Unclear if in splitting the range they actually run the samples three times (once with each range) instead of one run. If that's the case of three runs, the conclusion that one detects more proteins in total from the three runs seems to be obvious. And there is a disadvantage of increase in the time of analysis, as each sample needs to be run thrice.

Fig 2. Would be useful to see a similar figure for peptides (now only Protein groups is shown).

In Fig. 2.a, it should be indicated what the 2 different colors represent. In the legend it is written that blue bar indicates common identification from the triplicate experiment. So, can it be assumed the other color (grey?) represent what is not common in the replicates. If so, why there are these “not common” should be discussed. What impact could they have in the DIA method evaluation?

Section 3.3, line 262, How was “total protein intensities” obtained? Was it through library search? Or directDIA?

Section “3.3. Narrow precursor…quantification” should be Section 3.4.

Performing experiment three times is not good enough for evaluating reproducibility. It could be expressed as repeatability. To evaluate repeatability, SD should also be provided.

Fig. 5. Fold change and Q-value should be described. The order of OVAL and LYSC are reversed between Fig. 5a and 5b. Why? It can be understood that the values can vary between the two different methods but they should follow the same trend if the sample is the same. Or?

Discussion

Line 315, what are the authors referring to as “single shot analysis” and why so-called?

Could there be false positives, e.g. in the number of precursors, peptides and protein groups? The authors should discuss this and about other possible uncertainties in their outcome.

Author Response

Dear Reviewer,

Thank you for your insight comments/suggestions.

Please see the attached cover letter that include our responses to your comments.

Regards

Huoming

Reviewer 2 Report

The manuscript submitted by Zhang and Bensaddek described a method to increase the performance of DIA-MS in the identification and quantification of proteins by using a narrow acquisition window. They demonstrated the performance of this method with Arabidopsis sample with spike-in of known proteins with 34.7% more proteins identified and 59.4% more proteins quantified than in conventional DIA (cDIA) with a wide precursor range of 400-1200 m/z. This method would be useful for plant proteomics applications to quantify proteins with lower abundance.

 Minor issues

1 Figure 2 would be better to be split into two parts with one part showing the statistical difference between precursor windows and the other part showing the overlap with a venn diagram

2 Six known proteins were spiked into the samples. The authors should include in information of those proteins (peptides sequence, intensities from cDIA and cGPF-DIA, RSD, etc. ) as supplementary table.

Author Response

Comments and responses to Reviewer 2

The manuscript submitted by Zhang and Bensaddek described a method to increase the performance of DIA-MS in the identification and quantification of proteins by using a narrow acquisition window. They demonstrated the performance of this method with Arabidopsis sample with spike-in of known proteins with 34.7% more proteins identified and 59.4% more proteins quantified than in conventional DIA (cDIA) with a wide precursor range of 400-1200 m/z. This method would be useful for plant proteomics applications to quantify proteins with lower abundance.

            We thank the reviewer for his positive comment. We have implemented his/her suggestions as described below.

 Minor issues

1 Figure 2 would be better to be split into two parts with one part showing the statistical difference between precursor windows and the other part showing the overlap with a venn diagram

We included a venn diagram for the comparison between cDIA and cGPF-DIA in supplementary material 3. We prefer not to include venn diagram among triplicate runs as it is well known that there are always some proteins/peptides identified (up to 20% dependent on method and instrument) in one run but not in another in any LC-MS analysis. DIA-MS is better than DDA-MS in reproducibility in the identifying peptides in general.

2 Six known proteins were spiked into the samples. The authors should include in information of those proteins (peptides sequence, intensities from cDIA and cGPF-DIA, RSD, etc. ) as supplementary table.

We have now included the detailed identification and quantification of the 6 spike-in proteins in supplementary Table S6-8, and supplemental material 1-2, and cited them in main text (line 364-366): “The detailed information about their identification and quantitation are presented in Table S6 (protein information), Table S7-8 (peptide information) and illustrated in supplementary material.”

Round 2

Reviewer 1 Report

The authors have taken all my comments into consideration and revised the manuscript accordingly. I recommend this manuscript to be published.